# Engagement in meaningful activities post suicide loss: A scoping review

Monique Gill[1], Miranda Wu[1], Shania Pierre[1], Larine Joachim[1], Meera Premnazeer[1], Orianna Scali[1], Sakina J. Rizvi[2,3], Rebecca Renwick[1,4], Helene Polatajko[1,4☯], Jill I. Cameron[1,4☯]*

1 Rehabilitation Sciences Institute, Temerty Faculty of Medicine, University of Toronto, Toronto, Ontario, Canada, 2 Arthur Sommer Rotenberg Suicide and Depression Studies Program, St. Michael's Hospital, Toronto, Ontario, Canada, 3 Department of Psychiatry, University of Toronto, Temerty Faculty of Medicine, University of Toronto, Toronto, Ontario, Canada, 4 Department of Occupational Science & Occupational Therapy, Temerty Faculty of Medicine, University of Toronto, Toronto, Ontario, Canada

☯ These authors contributed equally to this work as they are co-supervising the first author's doctoral research.
* jill.cameron@utoronto.ca

## Abstract

### Background

Globally, more than 720,000 people die by suicide each year, leaving grieving individuals in their wake. Research indicates that individuals who lose a loved one to suicide face heightened risks for negative health outcomes. Recent studies show that taking part in meaningful activities can help protect health emphasizing the importance of exploring engagement in meaningful activities of everyday living among those bereaved. Currently, there has not been a review of the bereavement literature exploring the nature of, and extent to which, meaningful activities of everyday living are discussed.

### Objective

To explore the nature of, and extent to which the peer-reviewed, suicide bereavement literature addresses engagement in meaningful activities of everyday living.

### Methods

A scoping review following the Joanna Briggs Institute's framework was completed to summarize and map the literature. Four electronic databases were searched for two concepts: suicide and bereavement. Studies were screened using specific inclusion and exclusion criteria. Two independent reviewers completed title and abstract, and full text screening for each article. All conflicts were resolved through discussion or by a third reviewer. Data were charted, summarized and results were reported using the PRISMA Extension for Scoping Reviews.

**Data availability statement:** All relevant data are within the paper and its Supporting information file.

**Funding:** M. Gill is a PhD Candidate at the University of Toronto, and was supported by the Ontario Graduate Scholarship, Toronto Rehabilitation Institute Student Scholarship, Theresa and Miron Polatajko Graduate Award, Dalton Whitebread Scholarship Fund, the Dawson Family Scholarship, and the Peter Rappolt Family Scholarship for Research in Occupational Performance and Wellbeing in relation to this work. The funders did not have a role in study conceptualization, design, data collection, analysis, decision to publish, or preparation of the manuscript.

**Competing interests:** The authors have declared that no competing interests exist.

## Results

12372 studies were identified; 112 studies met inclusion criteria. Studies used qualitative (n = 90), quantitative (n = 10) and mixed (n = 12) methods. Findings indicate that the suicide bereavement literature discusses engagement in meaningful activities of everyday living using three main components: activities of everyday living, the engagement status of activities, and the meaning associated with activities.

## Discussion

While references to meaningful activities of everyday living appear in the bereavement literature, they typically are discussed within the background rather than central research aims. There is a need to bring this discussion to the forefront and view engagement in meaningful activities of everyday living as an important aspect of suicide bereavement.

## Introduction

Globally, more than 720,000 deaths occur due to suicide every year leaving families, friends, and communities to cope with this sudden loss [1]. Traditionally, suicide has been understood through clinical and biomedical lenses, commonly associated with mental health challenges and mental illness [2]. More recently, however, there has been a shift towards considering social, cultural and environmental perspectives, which highlight the role of broader contextual factors within understanding suicide [3–5]. Research also shows that being exposed to suicide increases the risk of suicidal ideations and/or attempts [6–8]. Therefore, addressing the needs of those bereaved is vital, as it can help prevent further loss and serve as an important approach to suicide prevention.

Within the bereavement literature, the term "survivors of suicide loss" is often used to describe those who are bereaving a suicide; therefore, this review will remain consistent with this terminology [9]. A recent survey completed by Cerel and colleagues estimated that each suicide death in the United States exposes approximately 135 individuals [10]. Those bereaved are particularly vulnerable to experiencing depression [11], post-traumatic stress disorder [12,13], and complicated grief [14]. Compared to other types of loss, they also face an increased risk of p1sychiatric admissions [9].

Building on recent literature that emphasizes the importance of contextual factors, survivors of suicide loss often encounter unique challenges when adjusting to life after loss [15]. These challenges include experiences of stigma leading to challenging emotions (e.g., shame, guilt, rejection, blame, etc.) [15], functional consequences (e.g., work, school, etc.) [16] and strained relationships before and following the death [15].

The documented health consequences and the unique challenges post suicide loss, contribute to difficulty adjusting to life [15,17]. This difficulty adjusting is often

called complicated grief, which has been shown to lead to challenges in engagement in meaningful activities of everyday living (MActEL) [18]. However, such challenges and difficulty engaging in MActEL can also occur after a suicide loss even without a diagnosis of complicated grief [16]. Currently, there is a need for further exploration on how MActEL are discussed within the suicide bereavement literature.

This scoping review was completed from an occupational perspective, defined as "a way of looking at or thinking about human doing" [19. p. 233]. This perspective is rooted in the occupational science and occupational therapy field where MActEL are also referred to as "occupations" [20]. MActEL or occupation is defined as any activity related to self-care, productivity, or leisure "that is performed with some consistency and regularity, that brings structure, and is given value and meaning by individuals and a culture" [21, p. 19]. The categorization of MActEL into self-care, productivity and leisure stems from the Canadian Occupational Performance Measure (COPM), an outcome measure designed to identify and evaluate an individual's self-perceived performance in MActEL [22]. The COPM is based on the Canadian Model of Occupational Performance and Engagement (CMOP-E), which provides the theoretical foundation for the COPM highlighting the interaction between the person, environment, and occupation [21,22].

Building on this perspective, Connor Schisler and Polatajko's work within occupational science speaks to the dynamic nature of engagement in MActEL as influenced by environmental factors while being mediated by the person [23]. They highlight that MActEL can be newly adopted, changed, abandoned, or remain constant in response to environmental and personal factors [23]. Thus, using an occupational perspective to explore the suicide bereavement literature and MActEL is in line with the shifting need identified in the field, specifically moving beyond biomedical and clinical perspectives, towards a focus on broader contextual factors to understand suicide and suicide bereavement.

The word "meaningful" is used intentionally in relation to activity, as a central assumption in occupational science and occupational therapy is that all occupations are inherently meaningful, with meaning assigned by an individual or culture [21]. Other fields have also explored the concept of meaning making. In psychology, for example, Neimeyer has extensively studied meaning making in his meaning reconstruction model, which proposes that reconstruction of meaning is essential following loss [24,25]. He describes meaning reconstruction as "a central process in grieving… the attempt to reaffirm or reconstruct a world of meaning that has been challenged by loss" [26]. As mentioned above, within this scoping review, MActEL refers to activities to which meaning is assigned by an individual or culture [21]. However, it is important to recognize that following loss, these meanings may shift or be reconstructed, as outlined in Neimeyer's model [24,25]. This idea aligns with the occupational science and occupational therapy models and frameworks adopted in this work (e.g., Connor Schisler and Polatajko's work on the dynamic nature of engagement) [21–23].

Bhullar, Sanford and Maple, as well as Miklin and colleagues emphasized the importance of focusing on meaning making and perceived individual impact following a suicide loss or suicide death exposure [27,28]. Previous research supports engagement in MActEL as an avenue to facilitating meaning in life, fulfilling basic psychological needs. and supporting physical and mental health recovery through fostering hope, identity, and connectedness [29–31]. Together, these findings further highlight the need for a literature review emphasizing the value of synthesizing and mapping the current suicide bereavement literature as it relates to MActEL.

Past reviews of the suicide bereavement literature have focused on: 1) attitudes towards suicide; 2) stigma; 3) the experience of suicide loss among health professionals; 4) comparison among types of losses; 5) ethics associated with studying survivors of suicide loss; 6) intervention effectiveness; 7) suicide exposure estimations; 8) the experience of survivors of suicide loss; and 9) supports and services for those bereaved [32–34]. A systematic review of the qualitative literature focusing on the bereavement process following suicide loss emphasized the important role that the meaning making process plays in adjusting to life post-loss, while highlighting three commonly discussed themes: 1) feelings following suicide loss; 2) the process of meaning making; and 3) the social context in which the previous two themes occur [35].

There is a clear need to prevent further loss by addressing the needs of those bereaved by suicide, particularly through a broader focus on the contextual factors that shape their experiences. The bereavement literature highlights the central

role of meaning making (or meaning reconstruction) in adapting to life after suicide loss, while occupational science and occupational therapy literature emphasizes that such meaning making can occur through engagement in MActEL. However, despite growing recognition of these links, no review has yet explored how MActEL are discussed within the suicide bereavement literature. Synthesizing this body of work is therefore essential to map what is currently known, identify critical gaps, and inform future directions. For this reason, a scoping review was undertaken to explore the nature of, and extent to which the peer-reviewed suicide bereavement literature addresses engagement in MActEL.

### Review questions

The primary research question for this scoping review was: What is the nature of and extent to which the peer-reviewed suicide loss and bereavement literature addresses engagement in MActEL?

A secondary question was: What key facilitators and/or barriers impacting engagement in meaningful activities post-loss are described in the literature?

### Method

A scoping review was completed in accordance with the Joanna Briggs Institute (JBI) guidance [36]. This review reports search results according to the Preferred Reporting Items for Systematic Reviews and Meta-Analyses Extension for Scoping Reviews (PRISMA-ScR) [37].

A scoping review methodology was deemed most appropriate as the aim was to identify, summarize and map the current literature discussing engagement in MActEL post-suicide loss with the goal of better understanding what has been studied and determining potential future research directions.

### Registration and published protocol

A search of MEDLINE, PsychINFO, and Open Science Registries was conducted before undertaking this scoping review (completed November 2022). No current or underway scoping reviews were identified on this topic. This review was registered with Open Science Framework Registries on January 23, 2023 (10.17605/OSF.IO/M2NES) and was conducted following an a priori published protocol [38]. All deviations from this published protocol are reported at the end of this paper [38].

### Eligibility criteria

The Population-Concept-Context (PCC) framework for scoping reviews assisted in identifying relevant studies and determining eligibility criteria [36]. The PCC framework for this scoping review is adapted from the original published protocol [38].

**Population.** This scoping review included studies completed among all age groups of individuals exposed to or impacted by suicide loss. Studies were excluded if they: 1) recruited a mixed sample of bereaved individuals (i.e., those bereaved by other mechanisms of death in addition to suicide loss); and 2) included survivors of suicide loss that had a professional relationship to the lost (e.g., teachers, therapists, healthcare workers, etc.).

**Concept.** Studies were included if they mentioned or discussed aspects of engagement in MActEL following a suicide loss. Studies were excluded if they solely examined short-term intervention participation (e.g., a study examining a 6-week Cognitive Behavioural Therapy program focused on anxiety symptom management) or research participation without discussing implications to MActEL following suicide loss.

**Context.** This scoping review included studies completed in any country or study setting. However, the review was limited to peer-reviewed literature published in English within the last 12 years. The last 12 years was set as a limit to ensure relevancy of results as related to our current world context, and to improve the feasibility of the synthesis.

**Types of sources.** This scoping review considered original qualitative, quantitative or mixed-method studies. Studies were excluded if they were categorized: grey literature, books, book chapters, book reviews, editorials, abstracts, unpublished studies, dissertations, organizational reports or knowledge syntheses. Regarding the latter, reference lists of all knowledge syntheses were searched to identify any additional studies; however, these knowledge syntheses were not included within this review to avoid duplicate representation of data.

## Search strategy

The search strategy aimed to locate all relevant peer-reviewed literature. Four databases, chosen in collaboration with two librarians at the University of Toronto, included: MEDLINE, PsychINFO CINAHL, and EMBASE. A three-step search strategy was utilized. Please see published protocol for a detailed description of the steps taken to create the search strategy used in this review [38].

Step one consisted of an initial limited search of MEDLINE and PsychINFO to identify appropriate search concepts and terms for the review aim. Following multiple consultations with two librarians at the University of Toronto, the search focused on two concepts: 1) suicide; and 2) bereavement/loss.

Next, the first 10 retrieved studies from MEDLINE and PsychINFO were examined to identify additional search terms. A second search was then done with all previously and newly identified keywords and index terms. Newly identified keywords and index terms included 1) suicide, completed; 2) kill oneself (searched as "kill* adj1 (onesel* or one-sel*)"); and 3) kill themselves (searched as "kill* adj1 themsel*"). This new search strategy was run across all chosen databases for this review (MEDLINE, PsychINFO, CINAHL and EMBASE). Again, the first 10 retrieved studies from each database were examined, however, no new keywords or index terms were identified.

Step two aimed to increase the comprehensiveness of the search strategy by using a novel cascaded search method to identify any missing and relevant search terms. This method employed a retrospective year-by-year search of the literature as described in detail in the published protocol [38].

This iterative process aimed at capturing any language change that may have occurred within the suicide bereavement literature. This was done until no new search terms were identified; this occurred after searching across three years (starting in 2022, ending in 2020; 2023 was not used as a search year as the initial search was completed one month into the year). Newly identified keywords included exposure to suicide and suicide loss (searched as "expos* adj4 suicid*" and "loss* adj4 suicid*", respectively). The final search strategy included all identified keywords and index terms and was adapted for each chosen database. The finalized search was completed on January 30, 2023, and an updated search was completed on January 6, 2025. The MEDLINE search strategy is provided in S1 Appendix.

Step three consisted of a reference list search of all knowledge synthesis studies that met the eligibility criteria listed above. This included 15 studies identified as one of the following: literature reviews, critical integrative reviews, rapid reviews, scoping reviews, systematic reviews, and integrative systematic reviews [39–53]. Studies referenced within these knowledge syntheses were cross-referenced with the existing search results and missing studies were added to ensure comprehensiveness of the search strategy.

## Evidence screening and selection

All identified studies were uploaded into Covidence, a screening and data extraction tool for knowledge synthesis projects [54], and duplicates were removed. A blinded pilot test was completed by five independent reviewers (MG, SP, MW, LJ, MP) to ensure inter-rater reliability achieving a minimum of 80% agreement [36]. The first 25 studies to appear on Covidence sorted for relevance were pilot tested during both the title and abstract screening and full text screening [36]. Reviewers then consulted the research team to discuss conflicts and possible areas of improvement in the study selection process. The research team consisted of health researchers well-versed in the completion of literature reviews [55–57].

**Title and abstract review.**  Each title and abstract were independently reviewed for inclusion by two members of the review team. Reviewers were not the same for each article and multiple reviewers were involved throughout the process (MG, SP, MW, LJ, MP). Conflicts regarding the inclusion of studies were managed through discussion among the larger research team or by a third reviewer. Studies deemed to meet inclusion criteria moved onto a full text review.

**Full text screen.**  Full text screen was completed by two members of the review team. Disagreements between reviewers at this stage of the selection process were resolved through discussion with the larger research team, or with a third reviewer. Reasons for exclusion of studies were recorded and reported during the full text screen. The results of the search and the study inclusion process are reported in full and presented in a PRISMA Extension for Scoping Reviews in Fig 1 [37].

## Data extraction

The research team collaborated to develop and use the data extraction tool. This extraction tool was pilot tested by four reviewers (MG, SP, MW, LJ) using 10 studies to ensure a minimum of 80% agreement [58]. This process contributed to consistency in data extraction, inter-rater reliability and assisted with refining the data extraction process. Data extraction for the remaining studies was independently completed by members of the research team (MG, SP, MW, LJ). Data extraction remained an iterative process, where additional useful descriptive data was discussed with the research team and added to the extraction form (see S2 and S3 Appendix for all extracted data).

The following details were extracted from all studies: 1) authors; 2) title of article; 3) publication year; 4) country/countries where data was collected; 5) name of journal; 6) corresponding author and affiliation; 7) methodology; 8) study design; 9) sample size; 10) participant characteristics (e.g., age, sex, gender, time since loss, and nature of relationship with individual lost); 11) aim of study; and 12) direct quotations discussing meaningful activities of everyday living.

## Quality appraisal of included studies

Following JBI recommendations for scoping reviews, a quality appraisal of included studies was not completed [36]. As the aim was to summarize the current state of the literature rather than provide practice recommendations, a quality appraisal of included studies was not deemed warranted [59].

## Synthesis of results

Data were summarized quantitatively using counts/percentages (e.g., study characteristics and participant characteristics) and qualitatively using inductive content analysis (e.g., direct quotes addressing meaningful activities of everyday living) [60,61].

**Inductive content analysis.**  Inductive content analysis was completed by MG using NVivo v14 [61,62]. First, quotations were reviewed to assist in familiarization with data. Next, content categories (i.e., big picture meaning units) were assigned to extracted quotations while keeping the research question in mind. This step was taken to assist in organizing the data. Initially only two preliminary content categories were identified: activities discussed, and engagement status. However, following full analysis of this data, content categories were revised to include activities of everyday living (AEL), engagement status, and associated meaning. Following this, round two of coding consisted of line-by-line coding. This was completed to create subcategories and fine-grained codes. The development of these fine-grained codes and subcategories was a rigorous process that involved consistent reflection and revision.

During inductive content analysis, previous research, theories and conceptual frameworks were also referenced to assist in organizing fine-grained codes and subcategories [61]. The COPM, an outcome measure based on the CMOP-E, as well as Connor Schisler and Polatajko's work on mediated change in activity were used to organize the data [21–23].

As previously mentioned, the COPM and Connor Schisler and Polatajko's work align with the occupational science and occupational therapy literature guiding this review. Both conceptualize performance and engagement in an activity as a

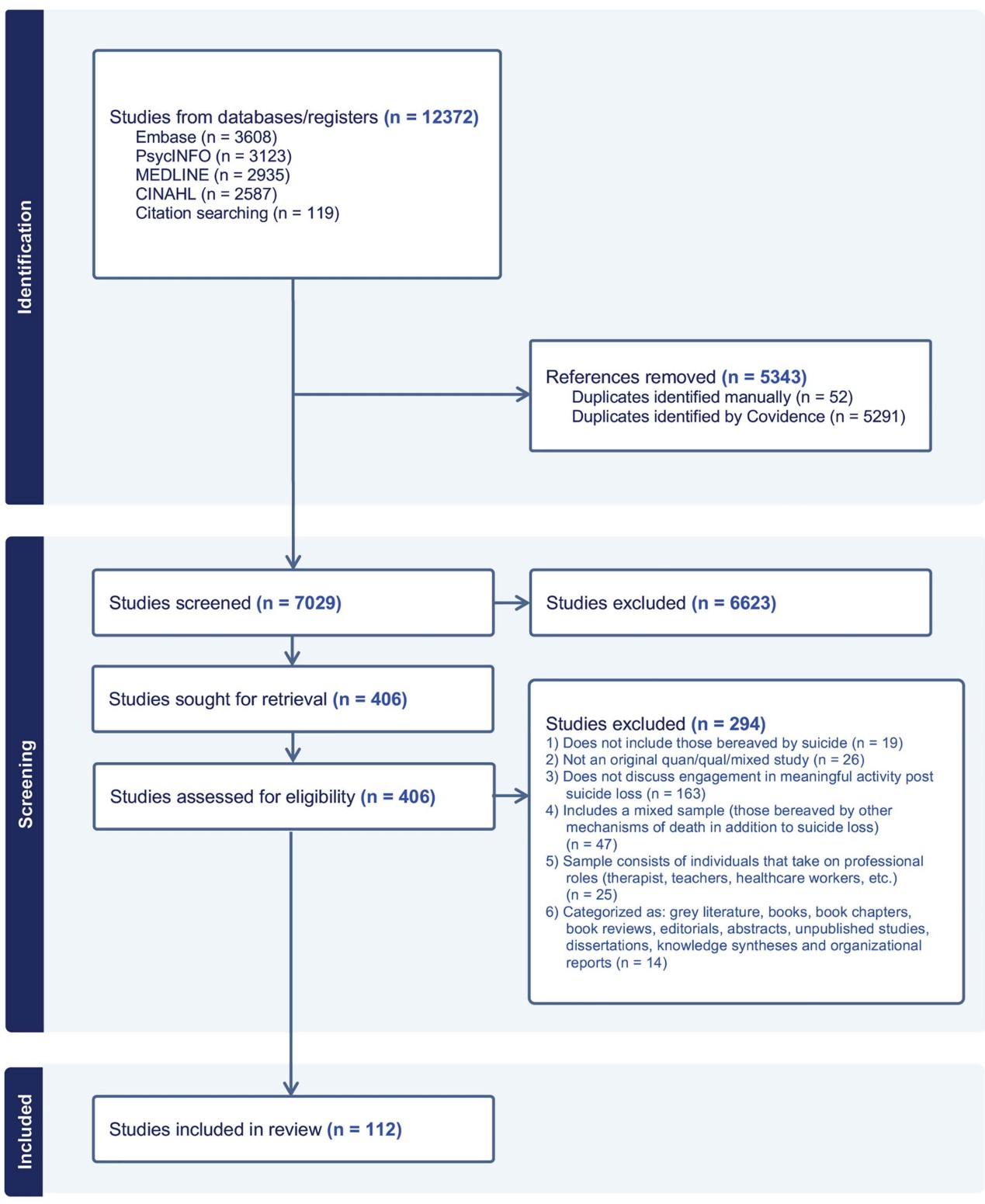

**Fig 1. PRISMA Extension for Scoping Review.**

dynamic interaction between person, occupation and environment [21–23]. The COPM categorizes activity using three domains: self-care, productivity and leisure [21,22], while Connor Shisler and Polatajko speak to the dynamic nature of engagement in activities as influenced by environmental factors and mediated by the person [23]. They categorize this as activity being the same as before (continued engagement), altered (change in method or frequency of engagement), abandoned (disengagement), or new (engagement in novel activity) [23].

Fine-grained codes and subcategories were created and grouped using the nomenclature provided by the COPM, and Connor Schisler and Polatajko's work [21–23]. To ensure consistency in the process of content analysis, MG defined each content category and subcategory. Table 1 outlines all definitions of content categories and their respective subcategories. Definitions are derived from the work above, in addition to current suicide bereavement literature, and occupational science and occupational therapy literature.

Finally, content categories were revised to ensure they were reflective of all subcategories and fine-grained codes. To synthesize data further, a matrix was created using all subcategories (see Table 2) with the aim of mapping concepts in relation to each other, to further understand how they were being discussed in the suicide bereavement literature. Using an iterative process of revisions, a mind map was also created throughout data analysis. The aim was to map the association of ideas on engagement in MActEL present within the literature [64].

**Table 1. Defining Content Categories and Subcategories.**

| Content Category | Definition | Subcategory | Definition |
|---|---|---|---|
| Activities of Everyday Living (AEL) | Any activity related to self-care, productivity, and leisure "that is performed with some consistency and regularity" [20, p. 19; 22]. | Self-care activities | Any activity necessary to care of oneself, get ready for the day, and move around the environment [22]. |
| | | Productivity activities | Any activity related to earning a living, maintaining a home and family, providing service to others, and/or developing one's capabilities [22]. |
| | | Leisure activities | Any activity an individual performs when freed from the obligation to be productive [22]. |
| | | Generic and/or unspecified activities | Generic: An activity or group of activities spoken to broadly (e.g., "routines and rituals related to the lost"). Unspecified: An activity or group of activities not clearly stated or defined (e.g., mentions of changes to "everyday activities" or to "day-to-day activities"). |
| Engagement Status | The status of an individual's participation in an activity, specifically stating how occupied or involved an individual is, potentially highlighting any changes in engagement or method of engagement [20]. | Change in frequency or method of engagement | Altering or modification to AEL through a change in the frequency or method of engagement [20,23]. |
| | | Continued engagement | Continued involvement and participation in an AEL [20,23]. |
| | | Disengagement | No longer involving oneself in/participating in or abandoning an AEL [20,23]. |
| | | Re-engagement | Resuming involvement and participation in an AEL previously engaged in [20,23]. |
| | | Engagement in a novel activity | New involvement and participation within novel AEL [20,23]. |
| Associated Meanings | This is referring to the meaning an individual is generating/generated or associating/associated with an activity. Meaning can be seen as both a process (meaning making) or an outcome (meaning made) [63]. Both types of meaning were considered during coding. | Surviving loss | Meaning making or meaning made to survive and/or endure the bereavement and grief associated with suicide loss [63]. |
| | | Managing and processing loss | Meaning making or meaning made to manage and process the bereavement and grief associated with suicide loss [63]. |
| | | Moving forward | Meaning making or meaning made to move forward after suicide loss [63]. |

**Table 2. Synthesis Across Content Categories Matrix.**

| | Associated Meaning | | |
| --- | --- | --- | --- |
| Engagement Status | Surviving Loss | Managing and Processing Loss | Moving Forward |
| **Change in Frequency or Method of Engagement** | Self-care | Self-care | Self-care |
| | Productivity | Productivity | Productivity |
| | Leisure | Leisure | Leisure |
| **Continued Engagement** | -- | Self-care | Self-care |
| | Productivity | Productivity | Productivity |
| | Leisure | Leisure | Leisure |
| **Disengagement** | Self-care | Self-care | Self-care |
| | Productivity | Productivity | Productivity |
| | Leisure | Leisure | Leisure |
| **Engagement in Novel Activity** | Self-care | Self-care | Self-care |
| | Productivity | Productivity | Productivity |
| | Leisure | Leisure | Leisure |
| **Re-engagement** | Self-care | Self-care | Self-care |
| | Productivity | Productivity | Productivity |
| | Leisure | Leisure | Leisure |

## Results

The search identified 12,253 studies before deduplication. 119 additional studies were identified following reference searching of all knowledge syntheses, totalling 12,372 studies before the removal of duplicates. 5343 duplicates were identified and removed. A total of 7029 studies went through title and abstract screening, of which 6623 were excluded. A total of 406 studies underwent full text screening, of which 294 were excluded. 112 studies met all study inclusion criteria and were included in this review [6,16,65–174] (Fig 1).

### Study characteristics

**Publication year.** Of the 112 studies in this review, 86 (77%) studies were published within the past six years between 2019–2024 [65–69,73,75,78–89,91,93–95,98–100,103,105–109,114,116,118,119,123,125,126,128,130,133–174]. As the final, updated search was run on January 6, 2025, no studies met inclusion criteria from the 2025 publishing year. The distribution of studies by the publication year in Fig 2 shows an increase in the discussion of MActEL within the suicide bereavement literature in the past 6 years.

**Countries where data were collected.** 49 of the 112 (44%) studies collected data from either the United States, United Kingdom or Australia, as seen in Fig 3 [6,16,65,67,69,71–75,81,82,84,85,89–92,94,96,103–106,108,109,111,114–117,119,120,128–130,132,134,135,138–140,149,161–163,167,168,170,174]. Most data originated from high-income countries, except for nine studies, which collected data from lower-middle income (India, Pakistan), and upper-middle income (Brazil, Turkey and China) countries [68,79,143–146,148,156,157,175].

**Methodology and study design.** Qualitative research was undertaken in 90 (80%) studies [6,16,65–69,71,72,74–84,86–108,110–119,122–126,128,129,131,133,135–146,148,150,151,155–158,160,161,164,165,168,169,171–173], with a variety of study designs identified. The most common qualitative study designs included interpretative phenomenology, narrative inquiry, and grounded theory. Mixed methods research was undertaken in 12 (11%) studies [73,109,120,121,127,130,134,152,162,163,170,174], while the remaining 10 (9%) studies completed quantitative research [70,85,132,147,149,153,154,159,166,167]. The most common quantitative designs included case studies/ reports, and cross-sectional survey designs.

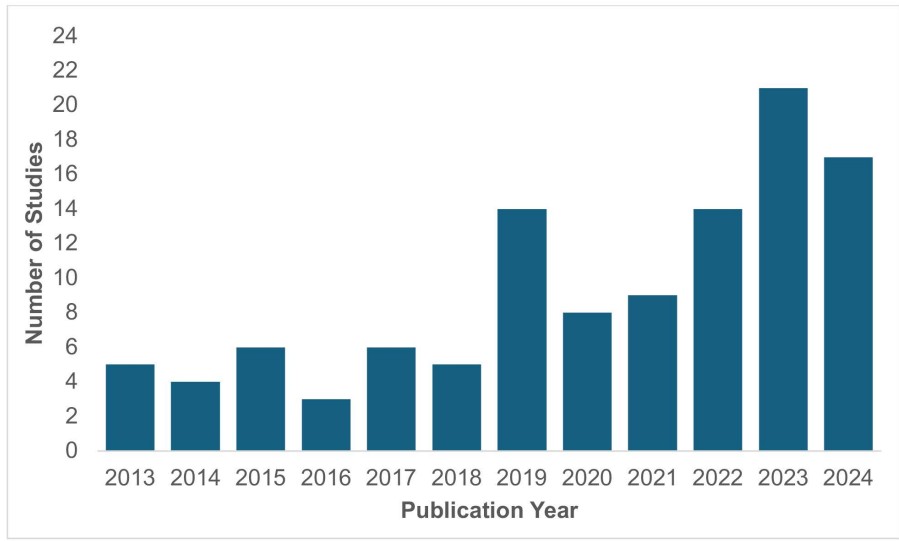

**Fig 2. Studies According to Publication Year.**

## Participant characteristics

**Participants recruited.** Of the 112 studies, 111 recruited participants [6,16,65–123,125–174]. One study examined online messages/posts and thus did not report a sample size [124]. Across all studies that recruited participants, the median number of participants recruited was 13. However, studies included did range from recruiting one participant to recruiting 3010 participants [6,16,65–123,125–174]. The median number of participants recruited for qualitative studies was 11 [6,16,65–69,71,72,74–84,86–108,110–119,122,123,125,126,128,129,131,133,135–146,148,150,151, 155–158,160,161,164,165,168,169,171–173], while the median for quantitative studies was 132 [70,85,132,147,149, 153,154,159,166,167].

**Age.** Of the 111 studies that recruited participants, 81 reported the age of the participants [6,16,65,67,69– 76,78–80,83,84,86,88–91,93–95,97–103,105–107,109,112,113,116,118–122,125–127,131–133,137–142,145– 152,155–158,160–164,166–173]. 60 studies, recruited a mixed sample which included children (0–17 years of age), young adults (18–29 years of age), adults (30–64 years of age) and older adults (65 + years of age) [6,16,67,69,70,73–76,78,80,83,84,88–91,93–95,98–101,105,107,112,113,116,119–122,127,131–133,137–142,146– 152,155,156,158,160,164,167–170,172]. 11 studies recruited solely adults [71,79,86,97,102,103,109,123,145,157,173], 7 studies recruited only young adults [65,72,106,161–163,166], and 3 studies recruited only children [118,126,171]. There were no studies that recruited only older adults.

**Sex and gender.** Of the 111 studies that recruited participants, 105 reported the sex of participants [6,16,65– 84,86,88–97,99–114,116–123,125–160,162–173]. 76 of the studies included a mixed sample of males and females [6,16,65–74,76,77,80,81,83,84,89–95,97,99–101,104–107,110–114,116,118,119,121,122,126,127,130,132– 137,139,140,143,144,146–150,152,155,156,158–160,162–165,167–170,173], 19 studies recruited only females [75,78, 82,88,102,103,117,125,128,129,131,138,141,142,151,153,157,166,172], and 10 studies recruited only males [79,86,96, 108,109,118,123,145,154,171].

The gender of participants was reported in 20 studies [80,83,103,109,137–139,141,142,145,146,153,158– 160,162,163,165–167]. The language used to conceptualize gender was inconsistent, including language describing

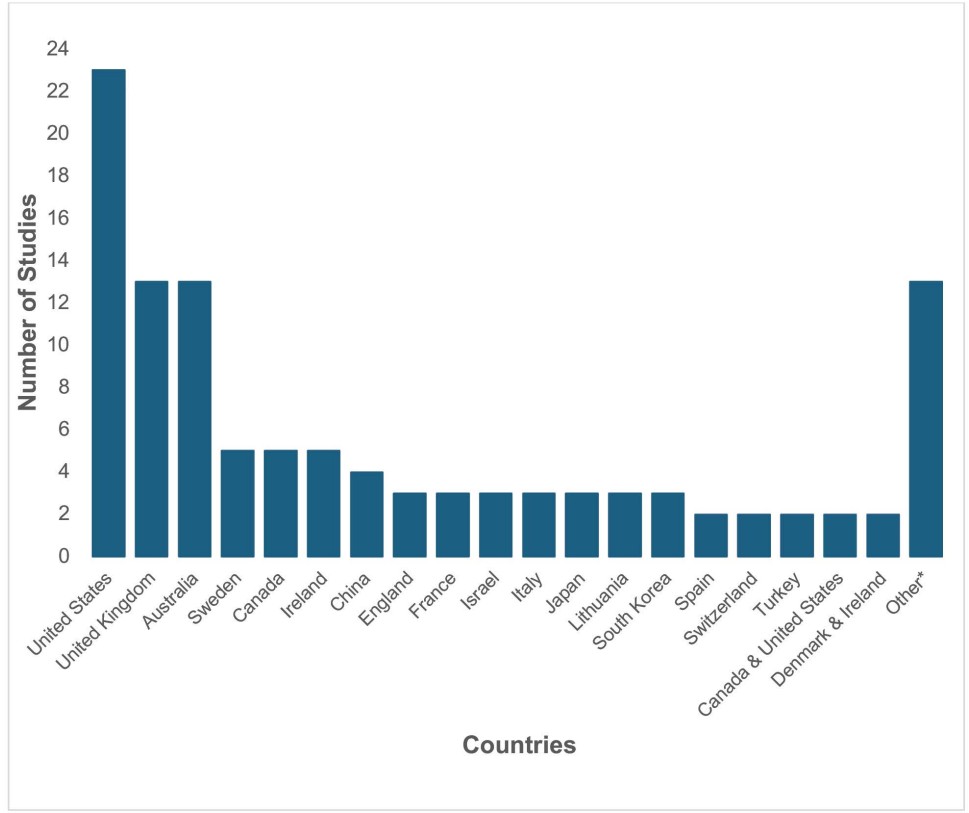

**Fig 3. Countries where Data was Collected Among Studies.** *For 13 included studies, data were collected in a unique country or unique group of countries. This included Brazil; Cyprus; Denmark; Germany; Iceland; India; Netherlands; Norway; Pakistan; Scotland; England and Wales; United States and Australia; or United States, United Kingdom, Australia and Canada.

gender identity (e.g., cisgender, transgender, man, woman) [103,109,139,141,142,145,146,153,160,165,166], language describing parental relations to a lost child (e.g., mother, father) [137,138,159], and language descriptive of sex to identify gender (e.g., male, female, non-binary, etc.) [80,83,158,162,163,167]. Due to inconsistencies in reporting and general limitations in data collected related to the gender of participants; these data were not analyzed.

**Time since loss.** 73 studies reported time since suicide loss among participants [6,16,65,67–72,74–78,80,85, 87–91,93–95,97–101,104,105,107,109,111–113,119–121,123,126,127,130,131,134,136–142,145–149,151–153, 155,157,160–164,166,168–170,173,174]. Of these, 19 studies reported time since loss as under five years [70,71,74,78,88,99,119,120,123,126,127,131,141,142,145,146,153,157,174]. 6 studies reported time since loss as five or more years ago [94,98,101,109,151,166]. 43 studies reported variability in the time since suicide loss [6,16,67–69, 72,75–77,80,87,89–91,93,95,97,100,104,105,107,111–113,121,134,136–140,148,149,152,155,160–164,168,169,173]. Five studies solely reported the mean time since suicide loss, ranging from 3.75–14 years since loss [65,85,130,147,170]. In summary, the studies presented a large amount of variability related to participants' time since suicide loss, as well as variability related to method of reporting.

**Relationship with individual(s) lost.** 106 studies reported the relationship between participants and individuals lost to suicide, [6,16,65,67–126,128–131,133,135,136,138–166,168–174]. Of the 106, 60 studies recruited participants with a variety of relationships to the lost person [6,16,67,69–71,74–77,79,80,83,85,87–91,93–95,97–100,105–107,111,

114–116,119,121,122,130,133,136,143,144,146–152,155,156,158,160,162–165,168–170,174], 13 studies focused on the suicide loss of a child [68,82,84,102–104,120,131,137,138,154,159,173], 10 studies explored the suicide loss of a parent [81,117,123–126,128,135,153,172], seven studies examined the loss of a sibling [65,101,112,113,118,139,166], seven studies focused on the loss of a partner or spouse [78,86,109,141,142,145,157], three studied the loss of a friend [72,73,129] and the remaining six studies explored losses of one of the following relationships: classmate, extended family member (uncle), close family member, colleague, peer, or relative [92,96,108,140,161,171].

**An inductive content analysis of meaningful activities of everyday living**

A total of 189 codes were generated from extracted quotations that addressed MActEL. These codes were reduced to 12 subcategories across three content categories AEL, engagement status, and associated meaning). Table 1 presents all definitions of content categories and their respective subcategories.

Below, we present the results of the inductive content analysis. A comprehensive table of codes linked to their subcategories and content categories is provided in S4 Appendix.

**Content category 1: Activities of everyday living.** Numerous references (n = 678) to a wide range of activities of everyday living (AEL) were identified across the 112 studies. Four subcategories were generated including self-care, productivity, leisure and generic and/or unspecified.

Self-care. There were 239 references made to self-care activities across 83 studies encompassing personal activities (e.g., eating, sleep, alcohol and drug use, risky and sexual behaviours), community management activities (e.g., driving and taking transit) and general self-care activities (e.g., meditation, attending long-term support groups and professional support, regular visits to mediums, religious activities, and general mention of self-care). Self-care activities were the second most frequently mentioned in the suicide bereavement literature, with only one fewer reference than productivity activities.

Productivity. There were 240 references made to productivity-related activities across 76 studies encompassing paid or unpaid work (e.g., caregiving or parenting, volunteer work (general), paid work, and mental health advocacy activities including film production, peer group facilitation, attending conferences, joining advocacy groups or organizations, research participation, and volunteer work (advocacy)), household management activities (e.g., housework, chores, and taking on role(s) of family members or a change in family life) and school or play activities (e.g., school or studies). Productivity activities were the most referenced group of AEL.

Leisure. Articles made 154 references to leisure activities across 59 studies encompassing quiet recreation activities (e.g., arts and crafts, baking, gaming, music, radio, television, photography, reading, sewing, and writing), active recreation activities (e.g., camping, gardening, travelling and physical activity including swimming, Tai Chi, exercise, walking, yoga and dancing), socialization (e.g., internet sites, social media use and generic social activities), and general leisure activities (activities to connect with lost one through hobbies enjoyed by lost one, enjoyed activity, hobbies, and shopping). Leisure activities present with the most variability in the specific activities engaged in by individuals within the literature.

Generic and/or Unspecified. In total, 47 references were made to generic and/or unspecified activities encompassing activities related to the creation of a new life, everyday activities, and routines and rituals related to the lost one.

**Content category 2: Engagement status.** Within the suicide bereavement and loss literature, AEL were discussed in relation to their specific engagement status. Five subcategories were generated: change in frequency or method of engagement, continued engagement, disengagement, re-engagement, and engagement in a novel activity.

Change in frequency or method of engagement. A change in frequency or method of engagement was referenced 270 times in 83 studies. It was the most discussed engagement status, with 31 unique codes associated with this subcategory.

*"'I was in my last year at University when the bereavement occurred and I became extremely stressed and depressed. It meant that trying to complete my dissertation project and prepare other coursework became extremely difficult. I was*

*supported by my partner though and was able to complete my degree satisfactorily (at one point I had contemplated dropping out).'"* [16. p. 8]

Continued engagement. In 19 studies, 38 references were made to continued engagement in AEL described with 12 unique codes. As an example,

*"Jon was 58 when he lost his nephew to suicide after a troubled childhood. 'My work is very public so it's something I can't hide from... (I had to) swallow and push on.'"*[89, p. 779]

Disengagement. In 37 studies, 79 references were made to disengagement in AEL described with 17 unique codes.

*"Another challenge involved stigmatizing attitudes and family silence. As one participant shared: 'I had to quit my full-time job 18 months after my son took his life because of the stigma in the workplace'"* [110, p. 157]

Re-engagement. This was the least discussed engagement status within suicide bereavement literature with 36 references across 17 studies. Re-engagement in an AEL was described with 22 unique codes.

*"Some participants stated that they occasionally needed to force themselves to take part in activities that they used to enjoy. It might have been difficult to accept that life would go on and that they could be happy but participating in activities was an essential action to move forward in the process of transformation."* [87, p. 300]

Engagement in a novel activity. This is the second most discussed engagement status within suicide bereavement literature with 264 references across 81 studies. Engagement in novel AEL was described with 33 unique codes.

*"Conversely, participant 3 said that "We loved each other. Like it was like no other siblings. There was no bad blood. We never fought." She went on to mention that she had taken up his hobbies – "I learnt how to play piano… he gets me out of my comfort zone. He always has and continues to."* [139, p. 10]

**Content category 3: Associated meaning.** Research within the suicide bereavement literature emphasized the associated meaning behind AEL across different engagement statuses. These associated meanings included surviving, managing and processing, and moving forward from an experience of suicide loss. Although a large majority of quotations pulled from included studies spoke to the associated meaning(s) of an activity, it is important to note that a small minority of quotations were unclear in their associated meanings and thus coded as "not descriptive enough/does not speak to meaning of activity".

Surviving loss. A total of 204 references were made to activities that were attributed to surviving and/or enduring the bereavement and grief associated with suicide loss across 74 studies. This associated meaning was present across all AEL and engagement status subcategories. As an example:

*"I worked the next day, because I couldn't bear to be at home. I told my boss I'd be late, because of what happened and everything, but I kind of needed to be at work. For me, that helped me. I still thought about it, it wasn't like I was putting it out of my mind, but it helped me have some distractions."* [86, p. 1284]

Managing and processing loss. A total of 225 references were made to activities attributed to managing and processing the bereavement and grief associated with suicide loss across 81 studies. This associated meaning was also present across all AEL and engagement status subcategories.

*"All of the survivors had reached out to other survivors of suicide via the Internet or through face-to-face support groups. Connecting with other suicide survivors via online support groups "gave me a purpose," said one participant who lost her brother five years ago"* [113, p. 333]

Moving forward. A total of 131 references were made to activities that were associated with moving forward after suicide loss across 58 studies. Again, this associated meaning was present across all AEL and engagement status subcategories. As an example:

*"Nadav talked at length about the various strategies he employs to maintain a close relationship with his deceased sister: every Friday, he puts flowers on his sister's grave; he has her car and drives it daily; he holds onto the notes and CDs he found in her car, keeps her driving license in his wallet and makes a conscious effort to involve his mother and others who knew her in discussions about her."* [101, p. 1111]

**Synthesis across content categories: Activities of everyday living, engagement status and associated meaning.** Table 2 maps the subcategories of each content category into a matrix. All types of AEL are represented when engagement statuses are cross-referenced with associated meanings, except for self-care. The AEL, self-care activities, are not discussed in relation to continued engagement to survive the experience of suicide loss. Self-care activities are present under the subcategories of continued engagement and activities to survive loss when examined separately. However, when looking for self-care activities that are coded as both continually engaged in, with an associated meaning of surviving loss, they are absent.

In addition, content categories with their respective subcategories are mapped in Fig 4 [64]. This map is a visual representation of the categories presented above. Within this mind map, AEL, engagement status and associated meaning are overlapping categories that make up the components of the discussion surrounding MActEL within the suicide bereavement literature. Each category is connected to the others with a bidirectional arrow, representing the reciprocal influence they have on each other. The subcategories within each category are also listed.

## Discussion

This scoping review is the first to explore the nature and extent to which the peer-reviewed suicide bereavement literature addresses engagement in MActEL. In total, 112 studies were found that provided information regarding engagement in MActEL over the past 12 years. Although the primary aims of the included studies varied, they nonetheless provided information on engagement in MActEL. Most studies used qualitative methodologies and originated from high-income countries. Following inductive content analysis, identified and synthesized studies yielded three key topics of discussion (i.e., content categories): AEL, engagement status, and associated meaning. These content categories, along with their 12 subcategories, capture the breadth and depth of discussion in relation to engagement in MActEL within the suicide bereavement literature. The sheer number of studies identified for inclusion in this review suggests that the bereavement process is, in part, shaped by and shapes engagement in MActEL. This indicates the importance of MActEL in shaping the suicide loss experience.

Previous literature has shown that engagement in MActEL can support physical and mental health recovery through fostering hope, identity, connectedness, while facilitating meaning in life and fulfilling basic psychological needs [29–31,176]. This concept is echoed in occupational science and occupational therapy literature through the assumptions held by those in the field in relation to occupation or MActEL [21]. These assumptions include 1) "occupation affects health and well-being", 2) "occupation organizes time and brings structure to living", 3) "occupation brings meaning to life", and 4) "occupations are idiosyncratic" [21, p. 21].

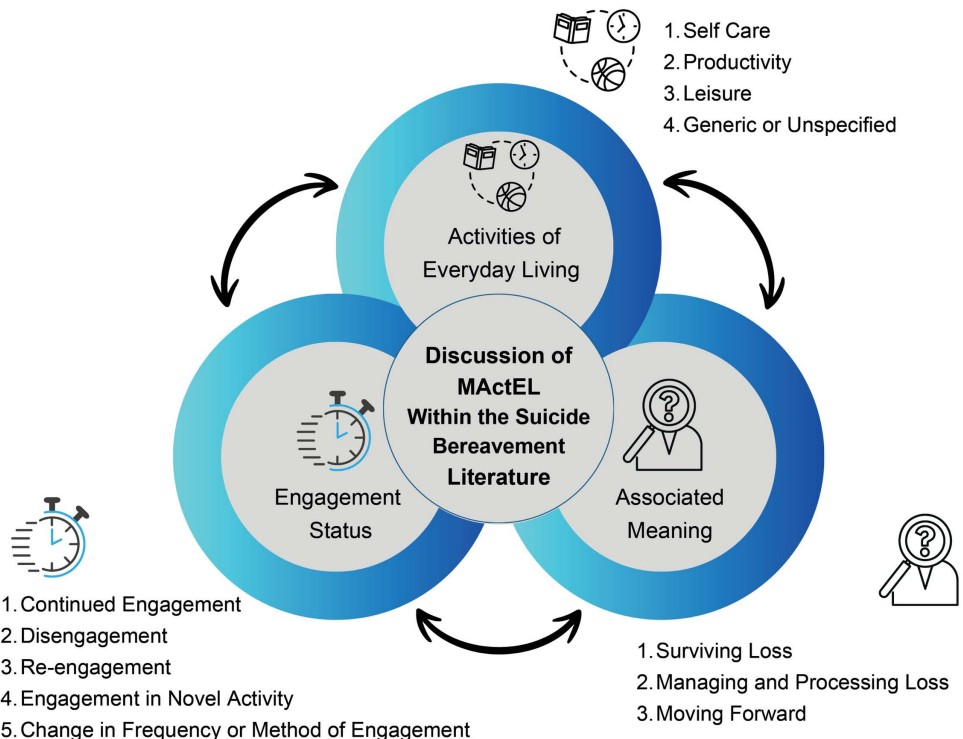

**Fig 4. Model of MActEL in Suicide Bereavement Literature.**

Results of this scoping review suggest similar roles for MActEL in suicide bereavement and call for focused research efforts on engagement in MActEL, as this has rarely been the primary aim of research. The findings underscore two key areas for discussion: 1) the need to view AEL, engagement status and associated meaning as interrelated concepts when examining suicide bereavement experiences; and 2) the importance of exploring engagement in MActEL as a component that shapes and is shaped by the suicide loss experience.

The findings of this review highlight that engagement in MActEL can be discussed through interrelated components of AEL, engagement status and associated meaning. While this review brings together these three concepts, the suicide bereavement literature has, almost exclusively focused only on associated meaning (referred to as meaning making or meaning reconstruction in the literature), specifically highlighting the need for meaning making of the loss experience as an essential step to process grief and move forward following suicide loss [24,177–179].

This understanding for the need of meaning making following the suicide loss experience stems from the research that highlights meaning making or meaning reconstruction as an essential step in moving through loss, specifically in existentialist frameworks within psychology and, thus, echoed within the bereavement literature [24–26,75,180]. Much of the suicide bereavement literature pulls from tenets of existentialism. Existentialism encompasses many perspectives but is often understood as the pursuit of living one's most authentic life while facing universal human challenges [181]. Within this framework, scholar Yalom identified four "givens" or ultimate challenges including death, freedom, isolation, and meaninglessness, which shape the human experience [180]. Building from this, much of suicide bereavement literature has focused on meaning making and meaning reconstruction as an essential step in processing grief and living with loss [24,177–179].

However, the findings of this scoping review emphasize that the discussion about associated meaning (meaning making or meaning reconstruction) commonly occurs in relation to two rarely recognized components: AEL and engagement status. Studies completed by Froese and colleagues, and Hybholt and colleagues suggest that a suicide loss experience changes the lives of individuals leading to a reconceptualization of daily life that is shaped by dynamic engagement in activities that create meaning or are meaningful to an individual [95,151]. The connection between AEL, engagement status and associated meaning post suicide loss has also been studied conceptually with a focus on engagement in leisure activities assisting individuals to re-establish a joyful, connected, discovered, composed and empowered life following suicide loss [182].

The findings of this scoping review also emphasize the need to conceptualize engagement in MActEL as a component that shapes and is shaped by life experiences. Occupational science and therapy theoretical frameworks and models like the Model of Human Occupation (MOHO), Occupational Adaption Theory, Person-Environment-Occupation Model, Canadian Model of Occupational Participation, and, the previously discussed, CMOP-E, all emphasize this understanding [20,183–187]. The idea of MActEL as a dynamic component has been explored in the occupational science and occupational therapy literature for decades, but it has rarely been included in other research arenas, including within suicide bereavement. Suicide bereavement literature often portrays engagement in MActEL as an endpoint rather than a means to an end [9,188]. The physical, mental, and socio-emotional consequences of suicide loss are cited as key factors affecting engagement in MActEL without outlining a reciprocal relationship [9,188]. This literature also does not identify engagement in MActEL as shaping or being shaped by life experiences as is highlighted in the occupational science and occupational therapy literature [29,176]. Only a few suicide bereavement studies have begun to identify the reciprocal relationship between engagement in MActEL and experiences of suicide bereavement. This is often done through the conceptualization of MActEL as interventions for those bereaved, highlighting the use of activity (e.g., creating a quilt) to work through grief reactions [189,190].

In addition to the need to focus on the reciprocal relationship between the suicide loss experience and engagement in MActEL, further research is needed to explore the socio-cultural, systemic, and individual (e.g., gender, socioeconomic status, etc.) contextual factors that shape this engagement. Much of the literature included in this review originates from high-income, Western contexts, where individualized coping and work-related re-engagement are often emphasized [191,192]. In contrast, in collectivist cultures, such as those in parts of Asia and South America, family obligations, rituals, and stigma may act as barriers or structured opportunities for MActEL following loss [143]. From a systemic perspective, concerns such as the criminalization of suicide and the consequential stigma, as well as the lack of public health measures addressing contributing factors can also influence whether and how bereaved individuals re-engage in MActEL [193,194]. Suicide remains criminalized in more than 20 countries worldwide, disproportionately affecting people in low- and middle-income contexts where stigma and fear of legal repercussions limit access to support [194,195]. Limited availability of culturally safe, affordable resources and uneven distribution of mental health services further compound these inequities [196]. At the individual level, gendered expectations, socio-economic status, inequities, among other factors may shape MActEL engaged in post suicide loss. Ultimately, these dynamics highlight the need for future research to capture more diverse cultural perspectives and to examine how intersecting identities influence engagement in MActEL following suicide loss globally.

Finally, it is important to note that following the completion of this scoping review, it was recognized that engagement in MActEL is not discussed using the terminology of "facilitators" and "barriers" within the suicide bereavement literature. Therefore, the secondary question (what key facilitators and/or barriers impacting engagement in meaningful activities post-loss are described in the literature?) could not be answered directly. However, this review did find that the engagement status of an AEL is closely linked with the meaning associated by an individual, noting a reciprocal relationship among these three components. This relationship ultimately shapes engagement in MActEL within the suicide

bereavement context. Future research could identify potential facilitators and barriers to engagement in MActEL by exploring the components of this reciprocal relationship in further detail, specifically studying engagement status, AEL and associated meaning as separate components.

## Limitations and future directions

This scoping review has some limitations. Generalizability of study results was impacted by the following factors: 1) only English language studies were included potentially excluding insights from studies in different languages and reflecting different cultural backgrounds; 2) studies were also identified as originating from higher income countries; and 3) the search excluded studies that recruited a mixed sample of bereaved individuals (i.e., those bereaved by other mechanisms of death in addition to suicide loss) and survivors of suicide loss that had a professional relationship to the lost (e.g., teachers, therapists, healthcare workers, etc.). In addition, the search excluded work categorized as grey literature, books, book chapters, book reviews, editorials, abstracts, unpublished studies, dissertations, organizational reports or knowledge syntheses. Studies were also excluded if they focused on short-term intervention participation. This may have also excluded potential discussion of engagement in MActEL.

Future research should aim to 1) explore the suicide bereavement literature on engagement in MActEL as interventions to improve health outcomes; 2) examine the discussion of engagement in MActEL outside of peer-reviewed literature (e.g., within grey literature) as well as within other socio-cultural contexts; and 3) study how engagement in MActEL may influence the suicide loss experience.

## Conclusions

This scoping review is the first to explore the nature of and extent to which the peer-reviewed suicide bereavement and loss literature addresses engagement in MActEL. The findings reveal that engagement in MActEL is rarely the primary focus of studies. This discussion occurs most often within qualitative research and revolves around three interrelated components: 1) AEL, 2) engagement status, and 3) associated meaning. This review identifies a need for focused research that brings the discussion on the engagement in MActEL following suicide loss and its implications on the suicide bereavement process into the foreground.

## Changes to the published protocol

A scoping review protocol paper was published prior to the completion of this review [38]. Several changes were made to the original protocol to ensure alignment with the most up-to-date research guidelines for scoping reviews [36]. These changes are listed in Table 3.

## Supporting information

**S1 Appendix. Search Strategy for MEDLINE via Ovid.**
(PDF)

**S2 Appendix. Data Extraction Table.**
(PDF)

**S3 Appendix. Data Extraction Table: Direct Quotes for Inductive Content Analysis.**
(PDF)

**S4 Appendix. Comprehensive Coding Table for Inductive Content Analysis.**
(PDF)

**Table 3. Changes to the Published Protocol.**

| Change Area | Plan Outlined in Published Protocol | Changes Made |
|---|---|---|
| Methods | "This scoping review will be guided by scoping review stages identified by Arksey and O'Malley [197] and updated by Levac and colleagues [58], while being guided by the methodology outlined by Joanna Briggs Institute (JBI). [36]" [38, p. 4] | This review used only recent recommendations by the JBI [36] as they encompass all previous recommendations outlined by Arksey and O'Malley [197] and Levac and colleagues [58]. The decision to use one all-encompassing methodological framework was made to ensure alignment with the most recent scoping review recommendations. |
| Population Concept Context (PCC) Framework: *Population* | "The planned scoping review will include individuals that fall into any of the categories described by Cerel and colleagues [198]…" [38, p. 2] "1) suicide exposed; 2) suicide affected; 3) suicide bereaved, short term; and 4) suicide bereaved, long term [198]. Due to limited literature in the field, all age groups will be included." [38, p. 4] | The population parameter of the PCC framework was altered to all individuals exposed to or impacted by suicide loss, moving away from the use of the Continuum of Survivorship [198]. This decision was made as recent literature cautions against the use of degree of closeness to the individual lost as a method of categorizing suicide loss impact (including distress reactions and support needs) [27,28]. Studies have seen survivors of suicide loss who fall outside of these survivorship profiles [27]; therefore, to ensure comprehensiveness of this review, all studies discussing individuals exposed to or impacted by suicide loss were screened. |
| Population Concept Context (PCC) Framework: *Context* | "… this scoping review will be limited to suicide loss literature after 1969." [38, p. 5] | The search was limited to the last 12 years (2013–2025) rather than since 1969 to ensure only relevant research was being summarized as related to the current research context, in addition to improving the feasibility of this scoping review. |
| Inclusion Criteria and Reference List Searching | "Inclusion criteria. To be included studies must: 1) be in a published peer-reviewed journal article; 2) be considered an original literature review or a qualitative/quantitative/mixed methods study, written in English…" [38, p. 4] "Additional studies will be identified by reviewing the reference lists of included articles and previous review articles following the full text review." [38, p. 5] | Knowledge synthesis and literature review studies were removed from inclusion criteria and added to exclusion criteria as including them within the scoping review introduces duplicate consideration of studies. Reference list searching was only completed for knowledge synthesis studies that were retrieved during the initial and updated search and met inclusion criteria during full text screening stage. Once references were screened, knowledge synthesis studies were removed from included studies. |
| Exclusion Criteria | "Exclusion criteria. This review will not include grey literature, books, book chapters, unpublished studies, dissertations, abstracts, and reports. In addition to barriers such as time and resource constraints, these criteria will enable the exclusion of non-peer reviewed work which will assist in ensuring studies included within the review will be of better methodological quality." [38, p. 5] | An exclusion criterion was added: Studies were excluded if individuals bereaved by suicide took on a professional role concerning the lost one (teachers bereaving a student, therapists bereaving a client, healthcare workers bereaving a patient, etc.). This was done as unique bereavement experiences are often voiced within this population, such as a change in professional identity and contemplating competence, among others [199,200]. |
| Quality Appraisal | "The quality of included studies will be reported using the appropriate critical appraisal tool…" [38, p. 6] | Following recommendations of the JBI, quality appraisal of included papers was not completed. As the goal of a scoping review is to summarize available research on a topic rather than critically appraise research or provide practice recommendations, therefore, a quality appraisal was not completed [59]. |
| Data Analysis | "Steps outlined by Elo and Kyngäs for inductive content analysis of qualitative data will be followed [60]." [38, p. 7] | Recommendations by Elo and Kyngäs, as well as, Vears and Gillam were be followed for inductive content analysis [60,61]. This decision was made to ensure alignment with updated recommendations for inductive content analysis methodology. |

# Author contributions

**Conceptualization:** Monique Gill, Sakina J. Rizvi, Helene Polatajko, Jill I. Cameron.

**Data curation:** Monique Gill, Miranda Wu, Shania Pierre, Larine Joachim, Meera Premnazeer, Orianna Scali.

**Formal analysis:** Monique Gill, Miranda Wu, Shania Pierre, Larine Joachim, Helene Polatajko.

**Funding acquisition:** Monique Gill.

**Investigation:** Monique Gill.

**Methodology:** Monique Gill, Meera Premnazeer, Jill I. Cameron.

**Project administration:** Monique Gill.

**Supervision:** Sakina J. Rizvi, Rebecca Renwick, Helene Polatajko, Jill I. Cameron.

**Visualization:** Monique Gill.

**Writing – original draft:** Monique Gill.

**Writing – review & editing:** Monique Gill, Miranda Wu, Shania Pierre, Larine Joachim, Meera Premnazeer, Orianna Scali, Sakina J. Rizvi, Rebecca Renwick, Helene Polatajko, Jill I. Cameron.

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
