## [Decision Letter · Decision Letter 0]

8 Sep 2025

Dear Dr. Cameron,

Thank you for submitting your manuscript to PLOS ONE. After careful consideration, we feel that it has merit but does not fully meet PLOS ONE’s publication criteria as it currently stands. Therefore, we invite you to submit a revised version of the manuscript that addresses the points raised during the review process.

We look forward to receiving your revised manuscript.

Kind regards,

Sanja Batić Očovaj, PhD

Academic Editor

PLOS ONE

Journal Requirements:

https://pmc.ncbi.nlm.nih.gov/articles/PMC9408753/?

In your revision ensure you cite all your sources (including your own works), and quote or rephrase any duplicated text outside the methods section. Further consideration is dependent on these concerns being addressed.

4. We note that your Data Availability Statement is currently as follows: All relevant data are within the manuscript and in Supporting Information files.

Additional Editor Comments:

The discussion could be enriched with an in-depth analysis and comments on the results from the perspective of meaning-making. The existentialist framework can be applied in the interpretation of data.

In the sentence ”Studies used qualitative (80%), quantitative (9%) and mixed (11%) methods.”  use frequency instead of percentages since the number of studies is small.

Check the required format of the table for Plos One and increase the quality of figures.

Reviewers' comments:

Reviewer's Responses to Questions

**Comments to the Author**

1. Is the manuscript technically sound, and do the data support the conclusions?

Reviewer #1: Yes

Reviewer #2: Yes

2. Has the statistical analysis been performed appropriately and rigorously?

Reviewer #1: N/A

Reviewer #2: N/A

3. Have the authors made all data underlying the findings in their manuscript fully available?

Reviewer #1: Yes

Reviewer #2: Yes

4. Is the manuscript presented in an intelligible fashion and written in standard English?

Reviewer #1: Yes

Reviewer #2: Yes

Reviewer #1: Dear authors,

your manuscript is interesting, dealing with a topic that is sometimes underestimated: the life of suicide survivors. Moreover, also the focus on MActEL appear interesting.

Despite this, some amendments are necessary in order to improve the overall quality of the paper.

1. The introduction to MActEL was a bit scattered. Please rewrite it to make the introduction more fluent.

2. You acknowledge that most included studies are from high-income countries, but there is limited discussion on cultural factors in shaping MActEL post-suicide loss.

3. Consider discussing how sociocultural, gender, and systemic inequities influence engagement in meaningful activities during bereavement.

4. You note that their secondary question on barriers/facilitators could not be fully answered. While this is transparent, consider suggesting ways future studies could explicitly capture barriers/facilitators in relation to MActEL.

5. Figure are in low quality.

Reviewer #2: Thank you for the opportunity to review this manuscript. This is a scoping review on a topic of great interest and also great complexity. Deaths by suicide are devastating, cause enormous suffering in the families and survivors of these losses, and constitute a major global public health problem.

The manuscript, focuses on the review of publications on grief and suicide loss that address the participation of bereaved individuals in meaningful activities of daily living (MADL), is rigorous and exhaustive, and is based on numerous references, something unusual in reviews of this type, although its exploratory purpose justifies the number of references selected. I understand this to be a testament to the difficulty involved in extracting accurate information on the topic of study, as well as the complex and exhaustive procedure followed to select the documents on which the review was based.

The manuscript is well constructed and written, and its subject matter and methodological approach are relevant for publication. However, in my opinion, the submitted manuscript requires some improvements for publication.

1. The affiliations of two authors are missing: Shania Pierre and Larine Joachim.

2. The introduction does not sufficiently justify this scoping review of the published literature from the perspective of meaning-making after loss and the role of activities of daily living in that construction. I understand that it is essential to establish the scope of the activities of daily living of survivors of suicide loss as a destroyed life space that must be redefined.

In this sense, I believe that this perspective of meaning-making should be included more explicitly in the discussion.

3. In Table 2, you should complete the matrix with the "n" and "%" for each element to have all the information in the table.

4. In Figure 4, the model represented is not sufficiently explained in the text. What is specifically expressed by the intersection of the circles? What meaning do the bidirectional arrows have? Is everything connected and related to everything else?

5. Conclusions: The last conclusion lacks precision and concreteness. It seems more like a summary than conclusions based on the results.

I hope you can take these suggestions into account. It would be great to have your manuscript published soon.

**Do you want your identity to be public for this peer review?** For information about this choice, including consent withdrawal, please see our Privacy Policy

Reviewer #1: No

Reviewer #2: **Yes: ** M. Paz García-Caro

---

## [Author Response · Author response to Decision Letter 1]

15 Oct 2025

Dear Dr. Sanja Batić Očovaj, Academic Editor,

Thank you for the review of our manuscript, entitled "Engagement in meaningful activities post suicide loss: A scoping review" for publication in PLOS One and for all the helpful comments. We are pleased to submit an updated manuscript with tracked changes.

Below, please find responses to each of the editor’s / reviewers’ comments and our response.

1) Editor Comments and Responses

Issue raised: Please ensure that your manuscript meets PLOS ONE's style requirements, including those for file naming.

Response: The formatting of the manuscript has been changed throughout to meet PLOS ONE style requirements.

Specific changes: Changes to formatting of the manuscript: title page, abstract, headings, titles of table/figures, and body. Changes to supplementary information document titles.

Page #: 1-47, S1 Appendix, S2 Appendix, S3 Appendix, S4 Appendix

Issue raised: We noticed you have some minor occurrence of overlapping text with the following previous publication(s), which needs to be addressed: https://pmc.ncbi.nlm.nih.gov/articles/PMC9408753/?

In your revision ensure you cite all your sources (including your own works), and quote or rephrase any duplicated text outside the methods section. Further consideration is dependent on these concerns being addressed.

Response: Thank you for highlighting this in our manuscript, although any overlap was unintentional, we cross-referenced Kaspersen and colleagues (2022) with our own work and noted some similarities in sentence structure due to the use of a similar and very structured methodology (scoping review using JBI framework), as well as some conceptual similarities due to the small literature base on suicide bereavement (e.g., using similar references, common definitions and statistics). Changes have been made to the abstract, introduction and methods to address these. This paper is referenced in the introduction of this paper. In addition, the protocol paper (Gill et al., 2024) published related to this scoping review was revisited and compared side-by-side. Changes were made within the abstract and introduction, as well it has been cited more thoroughly throughout to assist in avoiding any potential self-plagiarism.

Specific changes related to Kaspersen and colleagues (2022) paper titled “Use of Health Services and Support Resources by Immediate Family Members Bereaved by Suicide: A Scoping Review”:

1. Abstract: “Globally, more than 720,000 people die by suicide each year, leaving grieving individuals in their wake. Research indicates that individuals who lose a loved one to suicide face heightened risks for negative health outcomes.”

“While references to meaningful activities of everyday living appear in the bereavement literature, they typically are discussed within the background rather than central research aims.”

2. Introduction: “Globally, more than 720,000 deaths occur due to suicide every year leaving families, friends, and communities to cope with this sudden loss [1].”

“Within the bereavement literature, the term “survivors of suicide loss” is often used to describe those who are bereaving a suicide; therefore, this review will remain consistent with this terminology [9]”

3. Methods: “A scoping review was completed in accordance with the Joanna Briggs Institute (JBI) guidance [26].”

Page #: Abstract: p. 2-3; Intro: p. 3; Methods: p. 7

Specific changes related to Gill and colleagues (2024) paper titled “Engagement in meaningful activities post suicide loss: A scoping review protocol”:

1. Abstract: “Recent studies show that taking part in meaningful activities can help protect health emphasizing the importance of exploring engagement in meaningful activities of everyday living among those bereaved. Currently, there has not been a review of the bereavement literature exploring the nature of, and extent to which, meaningful activities of everyday living are discussed.”

2. Introduction: The introduction has been rewritten. Please see document.

3. Methods: References made and added to credit original published protocol.

Page #: Abstract: p. 2-3, Intro: p. 3-6, Methods: p. 7-8

Issue raised: Please note that funding information should not appear in any section or other areas of your manuscript. We will only publish funding information present in the Funding Statement section of the online submission form. Please remove any funding-related text from the manuscript.

Response: All funding information has been removed from the manuscript.

Specific change: Deleted text - “Funding Source. M. Gill is a PhD Candidate at the University of Toronto, and was supported by the Ontario Graduate Scholarship, Toronto Rehabilitation Institute Student Scholarship, Theresa and Miron Polatajko Graduate Award, Dalton Whitebread Scholarship Fund, the Dawson Family Scholarship, and the Peter Rappolt Family Scholarship for Research in Occupational Performance and Wellbeing in relation to this work. The funders did not have a role in study conceptualization, design, data collection, analysis, decision to publish, or preparation of the manuscript.”

Page #: 33

Issue raised: We note that your Data Availability Statement is currently as follows: All relevant data are within the manuscript and in Supporting Information files.

Response: Confirming that, at this time, this submission contains all raw data required to replicate the results of this study.

Specific change: Addition of gender data to S2 Appendix - Data Extraction Table. Additional S3 Appendix – Data Extraction Table: Direct Quotes for Inductive Content Analysis added to Supplemental Information, previous S3 Appendix C relabelled to S4 Appendix.

Page #: S2 Appendix, S3 Appendix and S4 Appendix

Issue raised: The reviewers gave you suggestions that would improve your work. The introduction should give more information about MActEL, but also about the occupation theories that you mentioned later in the text.

Response: Thank you for this summary comment of reviewer feedback. This has been addressed in the introduction through the addition of three paragraphs discussing the occupational perspective, MActEL and occupational models / frameworks.

Specific change:

“The documented health consequences and the unique challenges post suicide loss, contribute to difficulty adjusting to life [15,17]. This difficulty adjusting is often called complicated grief, which has been shown to lead to challenges in engagement in meaningful activities of everyday living (MActEL) [18]. However, such challenges and difficulty engaging in MActEL can also occur after a suicide loss even without a diagnosis of complicated grief [19]. Currently, there is a need for further exploration on how MActEL are discussed within the suicide bereavement literature.

This scoping review was completed from an occupational perspective, defined as “a way of looking at or thinking about human doing” [20. p. 233]. This perspective is rooted in the occupational science and occupational therapy field where MActEL are also referred to as “occupations” [21]. MActEL or occupation is defined as any activity related to self-care, productivity, or leisure “that is performed with some consistency and regularity, that brings structure, and is given value and meaning by individuals and a culture” [22, p. 19]. The categorization of MActEL into self-care, productivity and leisure stems from the Canadian Occupational Performance Measure (COPM), an outcome measure designed to identify and evaluate an individual’s self-perceived performance in MActEL [23]. The COPM is based on the Canadian Model of Occupational Performance and Engagement (CMOP-E), which provides the theoretical foundation for the COPM highlighting the interaction between the person, environment, and occupation [22,23].

Building on this perspective, Connor Schisler and Polatajko’s work within occupational science speaks to the dynamic nature of engagement in MActEL as influenced by environmental factors while being mediated by the person [24]. They highlight that MActEL can be newly adopted, changed, abandoned, or remain constant in response to environmental and personal factors [24]. Thus, using an occupational perspective to explore the suicide bereavement literature and MActEL is in line with the shifting need identified in the field, specifically moving beyond biomedical and clinical perspectives, towards a focus on broader contextual factors to understand suicide and suicide bereavement.

The word “meaningful” is used intentionally in relation to activity, as a central assumption in occupational science and occupational therapy is that all occupations are inherently meaningful, with meaning assigned by an individual or culture [22]. Other fields have also explored the concept of meaning making. In psychology, for example, Neimeyer has extensively studied meaning making in his meaning reconstruction model, which proposes that reconstruction of meaning is essential following loss [25,26]. He describes meaning reconstruction as “a central process in grieving… the attempt to reaffirm or reconstruct a world of meaning that has been challenged by loss” [27]. As mentioned above, within this scoping review, MActEL refers to activities to which meaning is assigned by an individual or culture [22]. However, it is important to recognize that following loss, these meanings may shift or be reconstructed, as outlined in Neimeyer’s model [25,26]. This idea aligns with the occupational science and occupational therapy models and frameworks adopted in this work (e.g., Connor Schisler and Polatajko’s work on the dynamic nature of engagement) [22–24].

Bhullar, Sanford and Maple, as well as Miklin and colleagues emphasized the importance of focusing on meaning making and perceived individual impact following a suicide loss or suicide death exposure [28,29]. Previous research supports engagement in MActEL as an avenue to facilitating meaning in life, fulfilling basic psychological needs. and supporting physical and mental health recovery through fostering hope, identity, and connectedness [30–32]. Together, these findings further highlight the need for a literature review emphasizing the value of synthesizing and mapping the current suicide bereavement literature as it relates to MActEL.”

Page #: 4-6

Issue raised: The discussion could be enriched with an in-depth analysis and comments on the results from the perspective of meaning-making. The existentialist framework can be applied in the interpretation of data.

Response: Thank you for this great suggestion. Although this work stems from the occupational perspective from the occupational science and occupational therapy literature, a paragraph was added to speak to how suicide bereavement literature is rooted in the field of psychology and thus meaning making is often discussed in line with tenets of existentialism.

Specific change: “This understanding for the need of meaning making following the suicide loss experience stems from the research that highlights meaning making or meaning reconstruction as an essential step in moving through loss, specifically in existentialist frameworks within psychology and, thus, echoed within the bereavement literature [25–27,75,181]. Much of the suicide bereavement literature pulls from tenets of existentialism. Existentialism encompasses many perspectives but is often understood as the pursuit of living one’s most authentic life while facing universal human challenges [182]. Within this framework, scholar Yalom identified four “givens” or ultimate challenges including death, freedom, isolation, and meaninglessness, which shape the human experience [181]. Building from this, much of suicide bereavement literature has focused on meaning making and meaning reconstruction as an essential step in processing grief and living with loss [25,178–180].”

Page #: 27-28

Issue raised: In the sentence ”Studies used qualitative (80%), quantitative (9%) and mixed (11%) methods.” use frequency instead of percentages since the number of studies is small.

Response: This has been edited to reflect frequencies rather than percentages.

Specific change: “Studies used qualitative (n=90), quantitative (n=10) and mixed (n=12) methods.”

Page #: p. 2

Issue raised: Check the required format of the table for Plos One and increase the quality of figures.

Response: As per comment above, the formatting of the manuscript has been changed throughout to meet PLOS ONE style requirements.

Specific changes: Changes to formatting of the manuscript: title page, abstract, headings, titles of table/figures, and body. Changes to supplementary information document titles. All figures were run through PACE and NAAS software to ensure compatibility and figure titles removed. Quality was improved of all images.

Page #: 1-47, S1 Appendix, S2 Appendix, S3 Appendix, S4 Appendix, Fig 1-4

2) Reviewer 1 Comments and Responses

Issue raised: The introduction to MActEL was a bit scattered. Please rewrite it to make the introduction more fluent.

Response: The introduction, in its entirety, was rewritten to improve flow and better introduce MActEL.

Specific change: See introduction in manuscript.

Page #: 3-6

Issue raised: You acknowledge that most included studies are from high-income countries, but there is limited discussion on cultural factors in shaping MActEL post-suicide loss. Consider discussing how sociocultural, gender, and systemic inequities influence engagement in meaningful activities during bereavement.

Response: Thank you for this comment, as it is an important consideration. We have added a paragraph to address this.

Specific change: “In addition to the need to focus on the reciprocal relationship between the suicide loss experience and engagement in MActEL, further research is needed to explore the socio-cultural, systemic, and individual (e.g., gender, socioeconomic status, etc.) contextual factors that shape this engagement. Much of the literature included in this review originates from high-income, Western contexts, where individualized coping and work-related re-engagement are often emphasized [193,194]. In contrast, in collectivist cultures, such as those in parts of Asia and South America, family obligations, rituals, and stigma may act as barriers or structured opportunities for MActEL following loss [143]. From a systemic perspective, concerns such as the criminalization of suicide and the consequential stigma, as well as the lack of public health measures addressing contributing factors can also influence whether and how bereaved individuals re-engage in MActEL [195,196]. Suicide remains criminal

---

## [Decision Letter · Decision Letter 1]

29 Oct 2025

Engagement in meaningful activities post suicide loss: A scoping review

PONE-D-25-33075R1

Dear Dr. Cameron,

We’re pleased to inform you that your manuscript has been judged scientifically suitable for publication and will be formally accepted for publication once it meets all outstanding technical requirements.

Kind regards,

Sanja Batić Očovaj, PhD

Academic Editor

PLOS ONE

Additional Editor Comments (optional):

Reviewers' comments:

Reviewer's Responses to Questions

**Comments to the Author**

Reviewer #1: All comments have been addressed

Reviewer #2: All comments have been addressed

2. Is the manuscript technically sound, and do the data support the conclusions?

Reviewer #1: Yes

Reviewer #2: Yes

3. Has the statistical analysis been performed appropriately and rigorously?

Reviewer #1: Yes

Reviewer #2: N/A

4. Have the authors made all data underlying the findings in their manuscript fully available?

Reviewer #1: Yes

Reviewer #2: Yes

5. Is the manuscript presented in an intelligible fashion and written in standard English?

Reviewer #1: Yes

Reviewer #2: Yes

Reviewer #1: Dear authors,

thank you for your revisions. Now, the manuscript is more fluent and readable.

Now, it is suitable for publication.

Reviewer #2: The authors have satisfactorily addressed all the questions raised in the review, and the revised and modified version of the manuscript is therefore suitable for publication.

**Do you want your identity to be public for this peer review?** For information about this choice, including consent withdrawal, please see our Privacy Policy

Reviewer #1: **Yes: ** Francesco Maria Boccaccio

Reviewer #2: No

---

## [Editor Report · Acceptance letter]

PONE-D-25-33075R1

PLOS ONE

Dear Dr. Cameron,

I'm pleased to inform you that your manuscript has been deemed suitable for publication in PLOS ONE. Congratulations! Your manuscript is now being handed over to our production team.

Kind regards,

on behalf of

Dr. Sanja Batić Očovaj

Academic Editor

PLOS ONE